# Analysis of Chest X-ray for COVID-19 Diagnosis as a Use Case for an HPC-Enabled Data Analysis and Machine Learning Platform for Medical Diagnosis Support

**DOI:** 10.3390/diagnostics13030391

**Published:** 2023-01-20

**Authors:** Chadi Barakat, Marcel Aach, Andreas Schuppert, Sigurður Brynjólfsson, Sebastian Fritsch, Morris Riedel

**Affiliations:** 1School of Engineering and Natural Science, University of Iceland, 107 Reykjavik, Iceland; 2Jülich Supercomputing Centre, Forschungszentrum Jülich, 52428 Jülich, Germany; 3SMITH Consortium of the German Medical Informatics Initiative, 07747 Leipzig, Germany; 4Joint Research Centre for Computational Biomedicine, University Hospital RWTH Aachen, 52074 Aachen, Germany; 5Department of Intensive Care Medicine, University Hospital RWTH Aachen, 52074 Aachen, Germany

**Keywords:** deep learning, COVID-19, high-performance computing, image-based diagnostics, medical diagnosis support

## Abstract

The COVID-19 pandemic shed light on the need for quick diagnosis tools in healthcare, leading to the development of several algorithmic models for disease detection. Though these models are relatively easy to build, their training requires a lot of data, storage, and resources, which may not be available for use by medical institutions or could be beyond the skillset of the people who most need these tools. This paper describes a data analysis and machine learning platform that takes advantage of high-performance computing infrastructure for medical diagnosis support applications. This platform is validated by re-training a previously published deep learning model (COVID-Net) on new data, where it is shown that the performance of the model is improved through large-scale hyperparameter optimisation that uncovered optimal training parameter combinations. The per-class accuracy of the model, especially for COVID-19 and pneumonia, is higher when using the tuned hyperparameters (healthy: 96.5%; pneumonia: 61.5%; COVID-19: 78.9%) as opposed to parameters chosen through traditional methods (healthy: 93.6%; pneumonia: 46.1%; COVID-19: 76.3%). Furthermore, training speed-up analysis shows a major decrease in training time as resources increase, from 207 min using 1 node to 54 min when distributed over 32 nodes, but highlights the presence of a cut-off point where the communication overhead begins to affect performance. The developed platform is intended to provide the medical field with a technical environment for developing novel portable artificial-intelligence-based tools for diagnosis support.

## 1. Introduction

As the COVID-19 pandemic threatened to break down medical infrastructure all over the world, it became evident that effective and efficient methods of diagnosis are necessary in order to improve outcomes and save the lives of hospital patients [1]. Especially during the early phase of the pandemic, when antigen-based rapid tests were not yet available, there was an urgent need for alternative diagnostic procedures. The standard approach using reverse-transcription polymerase chain-reaction (RT-PCR) required a lot of time, trained staff, and laboratory capacity and showed, especially at the beginning of the pandemic, very heterogeneous accuracy [2,3]. Since pulmonary involvement in particular posed a risk to patients with COVID-19, it was reasonable to examine conventional chest-X-ray (CXR) images, which are a rapid and widely available diagnostic tool for COVID-19-specific changes [4]. Thus, early publications had already reported the presence of specific changes in thoracic imaging before a laboratory test yielded a positive result [5]. Focusing on readily available and inexpensive diagnostic procedures is especially meaningful as research predicts that such large-scale contagion events will happen at an increasing rate [6].

However, given the current advancements in high-performance computing (HPC) technology and the availability of commercial cloud computing (CC) resources to the general public, as well as large increases in online data storage and sharing capabilities, an increasing interest in machine learning (ML) and deep learning (DL) applications that put these resources to use in order to solve common problems can be observed [7,8,9]. Similarly, these techniques and resources are being employed towards extracting information from Big Data repositories that would otherwise require hundreds of researchers over several thousand hours [10,11]. More recently, the combination of HPC, Big Data, and ML have made headlines in the scientific community with the publication of two DL models, AlphaFold from DeepMind and RoseTTAFold from Baek et al., which match or even outperform existing methods for protein structure prediction [12,13].

It follows that several research groups have developed ML and DL methods for detecting COVID-19 from sonographic [14] and X-ray images of the thorax [15,16,17], or for predicting the mortality of COVID-19 patients from medical data [18], with all of the results highlighting how effective these models might be for quick triaging. In a similar application field, Rajaraman et al. merged several trained DL models to improve the diagnosis of pneumonia from CXR images with a higher success rate than conventional image recognition models [19]. Other researchers have made use of cutting-edge HPC resources, namely the Jülich Wizard for European Leadership Science (JUWELS) (https://www.fz-juelich.de/en/ias/jsc/systems/supercomputers/juwels (accessed on 19 December 2022)) cluster, one of Europe’s fastest supercomputers to train advanced DL networks on Big Data from different fields, thus highlighting the need to make use of modular supercomputing architecture (MSA) to advance the field of artificial intelligence (AI) [20]. Furthermore, advanced automated hyperparameter tuning methods such as KerasTuner (https://keras.io/keras_tuner/ (accessed on 19 December 2022)) and Ray Tune (https://docs.ray.io/en/latest/tune/index.html (accessed on 19 December 2022)) have been developed, which simplify the parameter search process needed to fine-tune the training of neural networks, thus yielding the best performing model without major interventions from ML researchers [21].

Application of the available HPC resources in the medical field, thus contributing to the analysis of medical data and a timely and precise diagnosis, has the potential to reduce the amount of stress that medical personnel are exposed to during their work [22,23]. Similarly, the medical field presents a fertile ground for setting up frameworks that can be easily loaded, modified, and deployed where needed to help mitigate the effects of future epidemics and pandemics [24]. In the present paper, these approaches are thus validated in the application of the COVID-Net developed by Wang et al. on newly obtained CXR images that were provided by healthcare partner E*HealthLine (EHL) as part of the European Open Science Cloud (EOSC) Fast-Track grants for COVID-19 research.

The work presented in this article describes the culmination of work performed towards setting up a platform within which medical data can be stored, cleaned, and analysed, and easily used to train ML and DL models [25,26]. The platform makes use of highly specialised hardware and software available at the Jülich Supercomputing Centre (JSC) to develop and train these models in the most efficient manner. These include firstly the DEEP and JUWELS supercomputing clusters, and the storage made available through the related projects. Advanced hyperparameter tuning methods are also used to fine-tune the models to produce the best results.

The following sections go into the details of (a) training COVID-Net on newly acquired data, (b) performing large-scale hyperparameter tuning on the model in order to extract the parameter combinations that produce the best trained models, and (c) re-training the model to highlight the improvement achieved in per-class accuracy for each of these combinations. Furthermore, resource scale-up is also performed in order to gauge the speed-up that can be achieved through the established platform.

Re-training the COVID-Net model in such a way serves as a preliminary proof-of-concept for the platform. Due to its easy adaptability to new use-cases and its portability on other academic or commercially available CC resources, this platform can support researchers in the medical field to create more complex models with better performance that would otherwise be impossible to develop due to a lack of computational resources and missing expertise in usage of HPC systems. Additionally, the models built and pre-trained within the platform rely on open-source data and software, making them easy to deploy on local machines in hospitals intensive care units (ICUs).

It is worth noting that several groups have applied hyperparameter optimisation to improve the results of DL-based COVID-19 diagnosis models [27,28,29]. However, comparison with these works cannot easily be undertaken, as the concept and specific innovation described in the present paper lies within scaling up the data storage, the model training, and the hyperparameter tuning processes through efficient use of HPC resources in order to cover more ground.

## 2. Materials and Methods

This section describes the hardware and software implemented within the developed data analysis and machine learning platform, as well as the methods and data through which the COVID-Net model, developed by Wang et al. [15], is re-trained on new data and its prediction performance is improved through large-scale hyperparameter tuning. Figure 1 presents a general overview of the re-training process and model improvement steps performed as part of the platform validation, and highlights how computationally expensive the hyperparameter tuning step is.

### 2.1. HPC Resources

In their presentation of a novel approach to build and organise HPC resources, Suarez et al. provide a thorough technical description of the hardware set up at JSC, with an emphasis on its modular aspects [30]. This is true in terms of the hardware dedicated to computation as well as that used for communication and for storage. In essence, the MSA allows for efficient scale-up as required by HPC researchers according to the tasks at hand.

The hardware is supported by the open-source scheduling software Simple Linux Utility for Resource Management (SLURM) (https://slurm.schedmd.com/(accessed on 19 December 2022)), which manages the workload over the available resources and leverages the scalability aspect of the modular system, but also reduces wasted computing time through intelligent prioritisation of tasks. Furthermore, aside from terminal access through SSH, users can directly access resources through an integrated Jupyter (https://jupyter-jsc.fz-juelich.de/ (accessed on 19 December 2022)) development environment, which can be adapted to the specific needs of the task at hand through pre-packaged data analytics and ML modules as well as personalised kernels and virtual environments.

#### 2.1.1. DEEP

The DEEP series of projects has been setting up the path towards exascale computing since 2016, focusing on scaling available HPC resources through boosters [31]. These projects have received funding granted by the European Commission under the Horizon 2020 program and have so far had three iterations under the titles “DEEP”, “DEEP-Extended Reach” (DEEP-ER), and “DEEP-Extreme Scale Technologies” (DEEP-EST). A fourth iteration upcoming as “DEEP-Software for Exascale Architectures” (DEEP-SEA) was launched in 2021 with the aim of delivering a standardised programming environment for exascale computing for the European HPC systems.

At the hardware level, DEEP-EST introduced the concept of MSA, making the cluster-booster architecture more attuned for data analytics tasks [32]. Accordingly, the system itself is divided into several modules, each sporting the necessary hardware for specific tasks (i.e., numerical data processing, image processing, hyperspectral image processing). These modules are presented in Table 1.

#### 2.1.2. JUWELS

The JUWELS supercomputer consists of two main parts: a cluster module and a booster module, commissioned in 2018 and 2020, respectively. The cluster module is a BullSequana X1000 system (https://atos.net/en/solutions/high-performance-computing-hpc/bullsequana-x-supercomputers/bullsequana-x1000 (accessed on 19 December 2022)) with 2583 nodes totalling 122,768 CPUs. Furthermore, several nodes are specialised for visualisation, large-memory, and accelerated computing tasks (https://apps.fz-juelich.de/jsc/hps/juwels/configuration.html (accessed on 19 December 2022)). The booster module, a Bullsequena XH2000 system (https://atos.net/wp-content/uploads/2020/07/BullSequanaXH2000_Features_Atos_supercomputers.pdf (accessed on 19 December 2022)), expands on the available computing power by adding a total of 940 nodes totalling 3744 GPUs.

In essence, the cluster module is intended for general-purpose computation tasks while the booster module allows for scalable computing, making large-scale simulation and visualisation tasks more possible [20]. By making use of the available high-speed network connections and available storage, the booster module has reached a peak performance of 73 petaflop per second. Kesselheim et al. validated its performance for large-scale AI research on several DL network training tasks across different fields. Their results and the recorded peak performance earned the JUWELS booster the top position on the fastest supercomputers in Europe in 2021 as well as the 7th spot on the international TOP500 list and the 3rd spot on the Green500 list.

For the purposes described in this manuscript, the development phase is performed on the DEEP-EST cluster and the usage of the JUWELS cluster and booster is reserved for large-scale production applications of the developed models.

### 2.2. Datasets

To validate the established platform, two separate datasets were used in order to train a pre-built classification model. The first dataset is the open-source COVIDx dataset (https://github.com/lindawangg/COVID-Net/blob/master/docs/COVIDx.md (accessed on 19 December 2022)), which was compiled by Wang et al. from a collection of open repositories as listed in Table 2 [15]. At the time of preparing the data, the most current version was COVIDx V8A. This dataset is subdivided into 3 main classes: Healthy, Non-COVID-19 Pneumonia, and COVID-19.

The second dataset was pre-compiled by industry partner EHL and made available through file transfer protocol (FTP). The dataset is subdivided into training and testing sets, each of which is further divided into different conditions including Healthy, Pneumonia, COVID-19, Atelectasis, and Cardiomegaly, among others. Further details about the dataset constitutions are presented in later sections of this manuscript, though it is worth mentioning that there was a considerable difference in the image resolutions between the two datasets as can be seen in Figure 2. Additionally, Table 3 describes the class distribution of images within each dataset.

Finally, in order to increase the robustness of the model to be re-trained, the two datasets were merged into a Fusion dataset, preserving the split structures shown in Table 4 and Table 5. The Fusion dataset represents the relatively heterogeneous data usually received from different medical institutions in special circumstances [33]. The applicability of the platform and its intended use on heterogeneous data represents one of the most important advantages.

#### 2.2.1. COVIDx Dataset

The process to obtain the COVIDx dataset is provided in detail as part of the COVID-Net Github (https://github.com/lindawangg/COVID-Net (accessed on 19 December 2022)) repository as it was compiled by Wang et al. [15]. The dataset was loaded into the online storage available at JSC and an analysis of the images was performed using the Open-Source Computer Vision Library (OpenCV) python package in order to verify that the dataset contains no duplicates or corruptions. The majority of the data provided in the COVIDx dataset are in the portable network graphics (PNG) image format. Table 4 presents the train-test split of the COVIDx dataset.

#### 2.2.2. EHL Dataset

The EHL dataset was made available through secure FTP and, similarly to the COVIDx dataset, loaded onto the online storage at JSC. The dataset is subdivided into several pulmonary and chest-related conditions, though for the purposes described in this manuscript solely the images within the Healthy, Non-COVID-19 Pneumonia, and COVID-19 directories were used. The remainder of the data will be used in a future transfer learning application of the available ML model.

After performing some verification steps on the data using OpenCV, it became evident that some images were duplicates of those available in the COVIDx dataset, which was traced back to the fact that one of the participating hospitals had made their data available as part of the Cohen dataset. These images were removed and the resulting distribution of data is presented in Table 5. The EHL dataset is made available as part of the European Open Science Cloud fast-track grant project and can be accessed online for research purposes (https://b2share.fz-juelich.de/records/aef5d3b8aa044485b9620b95b60c47a2 (accessed on 19 December 2022)). Evaluation of the trained models was performed using only the EHL dataset in order to verify these models’ ability to predict over the new data.

### 2.3. COVID-Net Model

The COVID-Net deep learning model was developed and released by Wang et al. in May of 2020 in response to the COVID-19 pandemic to screen patients for COVID-19 using chest radiographs [15]. The model follows the current DL standard for image analysis of using convolutional neural networks (CNNs) with intermittently varying kernel sizes, but expands on it by employing the residual architecture that was introduced by He et al. in their pioneering work on residual networks for object detection in images [34]. COVID-Net was built using TensorFlow (https://www.tensorflow.org/ (accessed on 19 December 2022)) version 1.13.

The initial approach with COVID-Net within the scope of this project involved running inference using the pre-trained model on both available datasets in order to highlight their differences, before moving forward with the re-training attempts, which also served the purpose of highlighting the potential speed-up that can be achieved using the available MSA.

#### 2.3.1. Model Selection

The Git repository for COVID-Net lists a number of models each with varying input image sizes and performance markers. At the time of performing this analysis, the best performing model was labelled “COVIDNet-CXR4-A”, which scales input images to a resolution of 480 × 480 pixels. Two other versions of the model exist that take inputs of lower resolution (224 × 224 pixels) with the best performing among them being “COVIDNet-CXR Large”. Both models are available for download from links in the repository.

Selecting the appropriate model for this application required an analysis of the resolutions of the available images, and since the majority of the images within the EHL dataset are below the threshold of 480 × 480 pixel resolution as can be seen in Figure 2, it became evident that the “COVIDNet-CXR Large” model would perform best. This decision is further supported by the initial inference results that will be presented below in Section 3, but follows the logic that down-sampling image data produces far less noise than up-sampling, which is more likely to generate artefacts by magnifying limited visual information.

#### 2.3.2. Model Training

The repository for COVID-Net provides scripts and terminal commands for training the network. These scripts define the training parameters (learning rate, number of epochs, batch size, location of the pre-defined network weights) and the location of the datasets for training and testing. Accordingly, the parameters are adapted to the updated datasets being used in this application, and a range is defined over which the training will be parallelised.

Additionally, the training script is updated in order to introduce the possibility of many concurrent parallelised training runs, thus making use of the available HPC resources. The initial approach for parallelised training was through performing a grid-search of pre-defined parameters to tune and iteratively populating a job-script that would then be submitted to the HPC scheduler. Instead, hyperparameter tuning is implemented, as described in the next subsection, which can streamline the parameter search and potentially uncover hyperparameter combinations that would otherwise have been missed. Finally, a set of parameters is selected to train the model with an increasing number of nodes, using the Horovod (https://horovod.ai/ (accessed on 19 December 2022)) distributed DL framework, in order to determine the extent to which training can be accelerated as more resources are made available.

### 2.4. Hyperparameter Tuning

Hyperparameters are parameters which influence an algorithm’s behaviour. These values are typically set by the user manually before the training of an algorithm. Choosing an optimal set of hyperparameters can significantly improve the performance of a model [35]. In order to easily find the best performing combination of parameters for training the COVID-Net model on the new and the combined datasets, the hyperparameter tuning library Tune, developed under the Ray framework, was employed [21,36]. This tuner takes a model and selected tunable parameters as input and performs an optimisation that highlights the combination of parameters that produces the best results according to a selected metric. Due to compatibility issues related to the earlier version of TensorFlow used in constructing COVID-Net, it was necessary to use version 0.6.2 of the Ray module.

The Ray framework employs schedulers that take advantage of parallel computing to scale up and speed up the task at hand; of these schedulers, population-based training (PBT), HyperBand, and Asynchronous HyperBand [37,38,39] are considered and compared to the default first-in, first-out (FIFO) scheduler. The comparison was performed by running the hyperparameter tuning process with each of the selected schedulers over the same parameter search space. The best-performing scheduler was selected based on runtime and the COVID-Net model’s performance when re-trained using the optimal parameter combination that the tuning process output.

## 3. Results

### 3.1. Pre-Optimisation Analysis

Running inference with COVID-Net on the available images highlighted the differences between the two datasets. The network performance on COVIDx was in line with the results published by the original authors. However, the images from EHL were more likely to be misclassified. In fact, the results presented in Figure 3a highlight a bias towards predicting COVID-19.

After re-training the network on a combination of the newly acquired images and the original COVIDx dataset, the results achieved are presented in Figure 3b, where classification accuracy is improved. In order to achieve these results, several training runs were performed in parallel where the class weights (CWs) were adjusted, as well as the learning rate (LR), the batch size, the COVID-19 percentages (CPs), and the number of training epochs. Through these training runs the range of these parameters that are tuned on a larger scale in the next step was narrowed down.

### 3.2. Hyperparameter Optimisation

The hyperparameter optimisation is performed on the DEEP-Extreme Scale Booster (ESB) partition, with 20 trials taking up 1 node each (see hardware configuration listed in Table 1). During these 20 trials the network is trained over 24 epochs, with each trial being assigned a different combination of the tunable parameters, in this case the COVID-19 percentage, the class weights, and the learning rate. The parameter values are chosen following a random uniform distribution in the case of the CWs and the CP, and a logarithmic uniform distribution for the LR. The selected schedulers distribute the tasks on the available nodes and in three of the four cases introduce further perturbations to the hyperparameters halfway through the training process. The specific experimental setup is further expanded in the below sections for each of the selected schedulers.

#### 3.2.1. First-In First-Out

The default scheduling algorithm for the Ray library, first-in first-out (FIFO), performs the basic scheduling task of distributing the trials over the available nodes and does not update the tunable parameters during the training process. It is employed here as a benchmark to gauge the performance of the other schedulers.

Running all the trials in parallel took a total of 402 min to complete, after which the best performing combination of parameters was an LR of 0.00013, CWs of 1 for healthy, 1.38745 for pneumonia, and 6.1508 for COVID-19, and a CP value of 0.289. These parameters were used to re-train COVID-Net over 50 epochs and the prediction performance of the model re-trained using these parameters is highlighted in Figure 4a. The trained model in this case is very capable of detecting COVID-19 infections in CXRs, but pneumonia cases are almost always diagnosed as healthy.

#### 3.2.2. HyperBand

The HyperBand scheduler is activated in this case halfway through the training process, at which point it begins stopping tasks that underperform. The trials required a total of 421 min to complete, at which point stopped trials were discarded while the best performing trial was selected based on the overall accuracy, loss, and run time.

Interestingly, several of the trials that presented high accuracy at the end of tuning did not perform well when trained, showing a complete bias towards predicting one of the three conditions. The prediction performance of a model trained on the selected best parameters of LR = 0.0006, CW = [1, 5.0312, 3.4151], and CP = 0.081 is presented as a heatmap in Figure 4b. The trained model was unable to provide certain predictions when exposed to the images from the test set even after training for 50 epochs. The highest overall prediction accuracy is for healthy patients, but that is still at 80%.

#### 3.2.3. Asynchronous HyperBand

Similarly to HyperBand, the Asynchronous HyperBand scheduler also implements early stopping, but does so while taking advantage of the available parallel processing power to distribute the tasks more efficiently.

Running the trials required a total of 422 min and the best performing model was chosen as having LR = 0.00012, CW = [1, 4.0981, 3.0387], and CP = 0.187. The outputs from the model trained on the best parameter combination from Asynchronous HyperBand are presented in Figure 4c. In this case, the generated parameters resulted in a trained model with improved results on the original re-trained COVID-Net presented in Figure 3b.

#### 3.2.4. Population-Based Training

The PBT scheduler introduces perturbations to selected parameters at a set time during the tuning process. This introduces an extra layer of randomness to the hyperparameter tuning and potentially uncovers new combinations from the different trials running in parallel. In this case PBT is tasked to begin perturbing the LR halfway through the total training time.

The trials ran for a total of 419 min and from the results LR = 0.00024, CW = [1, 9.9599, 9.4996], and CP = 0.346 were selected to be used for re-training COVID-Net, the predictive performance of which is presented in Figure 4d. Similarly to the results obtained in the Asynchronous HyperBand trial, this model also presented an improved performance in detecting pneumonia and COVID-19 cases although the “Healthy” prediction was reduced to 84%.

Figure 5 compares the prediction performance of the original re-trained COVID-Net model with that of models retrained using the best performing hyperparameters from the tuning process with Asynchronous HyperBand and PBT.

### 3.3. COVID-Net Re-Training

The Horovod framework was used to re-train the COVID-Net model based on parameters chosen from the previous results, while the resources available for training were iteratively increased. The graph presented in Figure 6a shows the change in training duration as more resources were made available.

The model trained significantly faster as the tasks were distributed among the increasing number of worker nodes. The time required to train over 25 epochs was reduced from 207 min on 1 node, to 54 min on 32 nodes. However, the rate of reduction decreased with resource increase as can be seen from the decreasing slope of Figure 6b. Ultimately, as the resources were increased to 64 nodes, the model training became slower and both curves switched to a positive slope, indicating that the cut-off point for speed-up had been reached.

## 4. Discussion

Through trial and error a set of parameters was selected to train the COVID-Net model on the Fusion dataset and the results obtained are shown in Figure 3b. In reality, several more parameters, including the batch size, the train-test split, the number of epochs, and freezing or unfreezing some layers from COVID-Net could have been tuned by hand in order to improve the results, but as the number of these parameters increases, so does the complexity of the optimisation problem. The results show that the model can be improved and highlight the fact that more effective tuning approaches are necessary.

Through four straightforward applications of a hyperparameter optimisation framework, it was possible to improve the predictive performance of COVID-Net on new data. The schedulers used for the optimisation took advantage of the available MSA and efficiently distributed the work over the available resources. In doing so, the framework was able to cover more ground and test more parameter combinations simultaneously in order to close in on the parameters with which the model would train more effectively. This process is not perfect, as can be seen from the results obtained from Hyperband, where the best-performing parameter combination yielded a model that underperformed, or through reducing the pneumonia class weights, the best performing parameters from the FIFO scheduler resulted in a model that was extremely good at finding COVID-19 patients, but completely incapable of predicting pneumonia. However, these results give insight into novel ways the parameters can be tuned and thus the model performance can be improved.

In the case of Asynchronous HyperBand and PBT, both resulting trained models performed more consistently than the original re-trained COVID-Net, with predictions trending towards true positives. The results also highlight the possibility of further improvement with longer training and further fine-tuning of the hyperparameters, both of which are made possible through the scale-up of the GPU resources on the compute clusters.

The reduction in training duration observed in Figure 6a is not infinite; in fact, as more nodes are recruited, the communication overhead between these nodes becomes more complex and more time-consuming, resulting in the flattening of the curve and ultimately the upward trend seen in Figure 6b. To counter this issue, it is important to understand the problem at hand and to recruit the appropriate hardware and software accordingly, while also performing many trials to pinpoint the cut-off at which training is the most efficient.

The work presented in this manuscript describes the large-scale re-training of COVID-Net as a use case to validate a modular medical diagnosis support platform built on an HPC infrastructure and taking advantage of novel and efficient ML algorithms. That is not to say that this work would not be possible without the specific HPC infrastructure used. In fact, the platform makes use of open-source software, making it easily portable onto commercially available cloud computing (CC) solutions. Similarly, the main aim is to develop the base infrastructure that takes advantage of the HPC resources to simplify the development of software that is lightweight enough to be easily deployed in most standard computers available in hospitals, making them a vital tool to support medical personnel.

Given that the medical field is regularly facing time-sensitive problems, this paper highlights the need for platforms that simplify access to cutting-edge resources for model training and development, and also for specially trained experts in the field of ML, data science, and HPC for medical applications, who would advise on applications, assist in setting up the problem solutions, and take part in the data analysis and the development of the diagnostic and treatment techniques of the future.

Finally, since the prototype platform described in this manuscript only used open-access data, there are no privacy risks and thus this issue was not addressed. As the platform moves towards production, and especially before dealing with restricted real-world data, its safety from outside threats will need to be assessed. Additionally, this process is still in its infancy and much work still needs to be done in order to test the robustness of this platform, and validate its performance in real-world use cases.

## 5. Conclusions

In the present manuscript, the re-training of a COVID-19 detection model was described as a use case through which an HPC-enabled data analysis and ML platform was validated. The MSA available at JSC, especially the scalable storage and computing resources, made it possible (1) to validate the performance of the COVID-Net model on the original COVIDx data as well as new data made available through research partners, (2) to perform large-scale hyperparameter tuning, through which the optimal training parameters for the model were uncovered, and (3) to re-train the model using the selected parameters and highlight the improvement that was achieved. Furthermore, the research also highlights the training speed-up that can be achieved using the platform.

The severity with which the COVID-19 pandemic struck worldwide, and research showing that such global phenomena may become more frequent, highlight the need for research platforms such as the one described in the present manuscript. These platforms would make use of highly efficient computing, communication, and storage technology, as well as open-source and interoperable software, and should be made available to assist the healthcare sector in order to simplify and accelerate the development of medical diagnosis support tools. This does not mean that medical institutions should be required to have access to HPC resources, which would put hospitals at a severe disadvantage, not only in developing countries. Rather, the models developed within these platforms ought to be more portable and easily implementable, while the communication channels between research institutions and medical centres ought to be strengthened, paving the way for effective medical and technological cooperation. Such platforms rely on the availability of data and the willingness of medical institutions to participate in the research, both of which are more likely to increase as the developed and validated models show beneficial effects in the field.

## Figures and Tables

**Figure 1 diagnostics-13-00391-f001:**
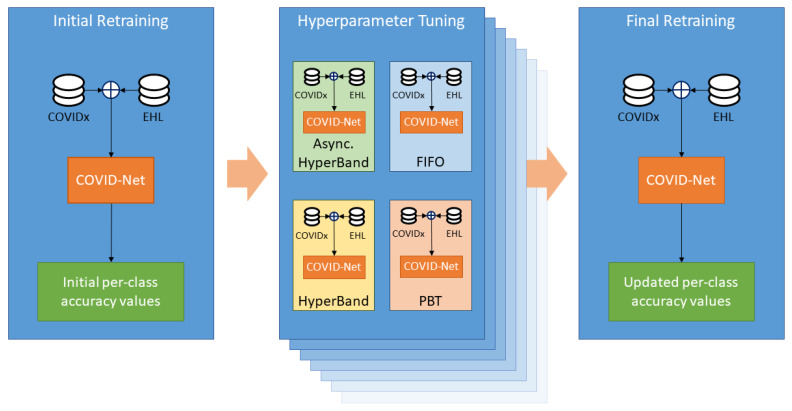
Block diagram representing the experimental process within the data analysis and machine learning platform. The different schedulers are represented as boxes within the hyperparameter tuning step. Due to the large amount of computations that it needs to perform, the hyperparameter tuning step requires significantly more resources than the remaining steps.

**Figure 2 diagnostics-13-00391-f002:**
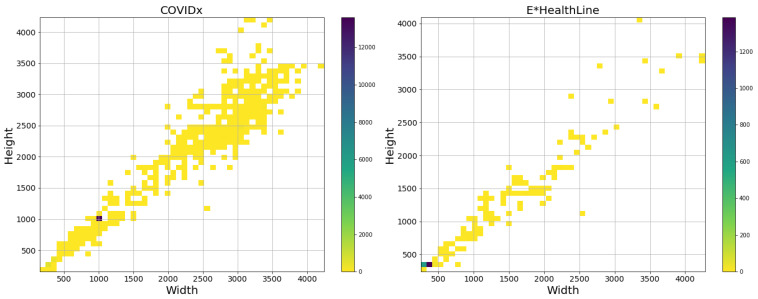
Range of image resolutions of the COVIDx (**left**) and EHL (**right**) datasets. A high concentration of images in the COVIDx dataset is centered around 1000 × 1000 pixels, but the majority of EHL images is below 480 × 480 pixels.

**Figure 3 diagnostics-13-00391-f003:**
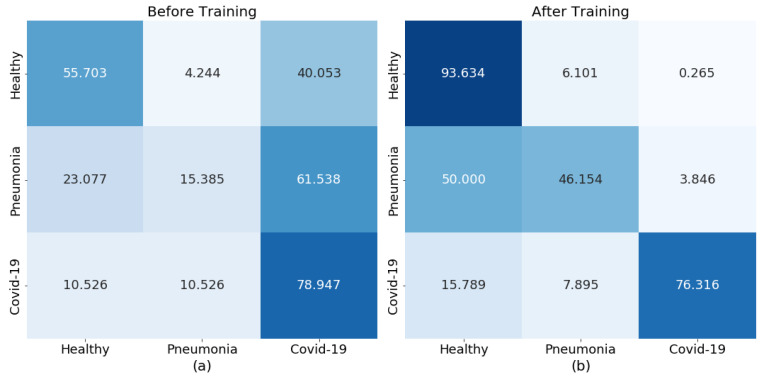
Prediction performance (in %) heatmaps for COVID-Net on the EHL dataset (**a**) before and (**b**) after initial re-training.

**Figure 4 diagnostics-13-00391-f004:**
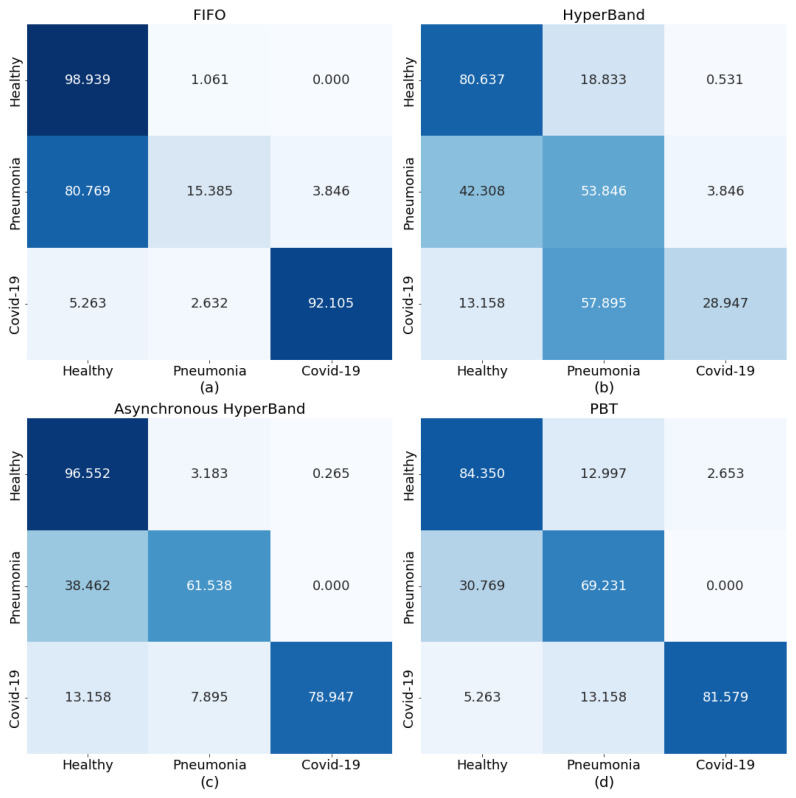
Prediction performance heatmaps for COVID-Net on the EHL dataset after re-training on the parameters chosen by (**a**) FIFO, (**b**) HyperBand, (**c**) Asynchronous HyperBand, and (**d**) PBT.

**Figure 5 diagnostics-13-00391-f005:**
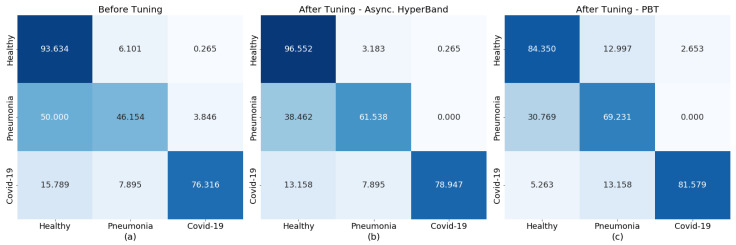
Comparison of trained COVID-Net prediction performance before (**a**) and after hyperparameter tuning with Asynchronous HyperBand (**b**) and PBT (**c**).

**Figure 6 diagnostics-13-00391-f006:**
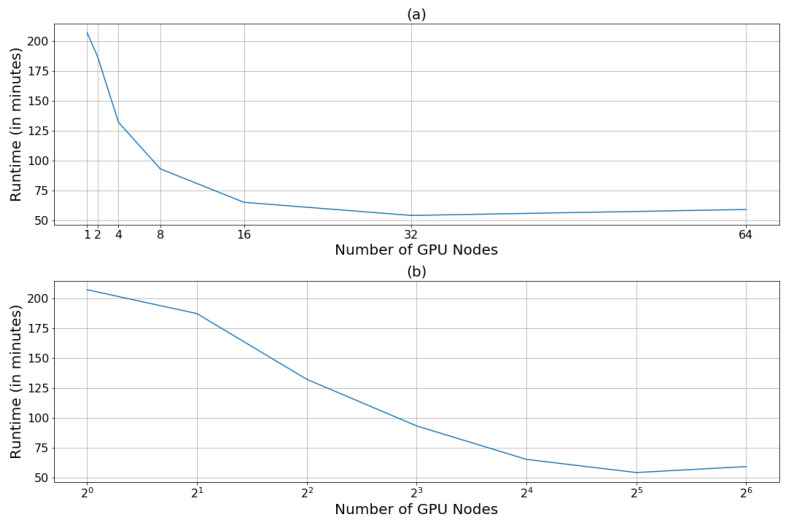
Training duration (in minutes) as more GPU nodes are recruited, (**a**) on a linear scale and (**b**) on a logarithmic scale.

**Table 1 diagnostics-13-00391-t001:** Partitions on the DEEP prototype.

Partition	Nodes	CPUs/Node	GPU
DEEP-Data Analytics Module	16	96	NVIDIA V100 + Intel Stratix10 FGPA
DEEP-Extreme Scale Booster	75	16	NVIDIA V100
DEEP-Cluster Module	50	48	n/a

**Table 2 diagnostics-13-00391-t002:** COVIDx V8A dataset sources.

Title	URL
Cohen	https://github.com/ieee8023/covid-chestxray-dataset
Figure 1	https://github.com/agchung/Figure1-COVID-chestxray-dataset
Actualmed	https://github.com/agchung/Actualmed-COVID-chestxray-dataset
Sirm	https://www.kaggle.com/tawsifurrahman/covid19-radiography-database/version/3
RSNA	https://www.kaggle.com/c/rsna-pneumonia-detection-challenge/data
RICORD	https://wiki.cancerimagingarchive.net/pages/viewpage.action?pageId=70230281

**Table 3 diagnostics-13-00391-t003:** Number of images within each dataset.

Dataset	Healthy	Non-COVID-19 Pneumonia	COVID-19
COVIDx	8066	5575	2358
EHL	1898	118	187
Fusion	9964	5693	2542

**Table 4 diagnostics-13-00391-t004:** COVIDx V8A dataset training and testing split.

Set	Healthy	Non-COVID-19 Pneumonia	COVID-19
Training	7966 (98.8%)	5475 (98.2%)	2158 (91.5%)
Testing	100 (1.2%)	100 (1.8%)	200 (8.5%)
Total	8066	5575	2358

**Table 5 diagnostics-13-00391-t005:** E*HealthLine dataset training and testing split.

Set	Healthy	Non-COVID-19 Pneumonia	COVID-19
Training	198 (10.4%)	21 (17.8%)	189 (65.4%)
Testing	1700 (89.6%)	97 (82.2%)	100 (34.6%)
Total	1898	118	289

## Data Availability

The COVIDx dataset is available online at https://www.kaggle.com/datasets/andyczhao/covidx-cxr2 (accessed on 19 December 2022). The EHL dataset is available online at https://b2share.fz-juelich.de/records/aef5d3b8aa044485b9620b95b60c47a2 (accessed on 19 December 2022). COVID-Net is available at https://github.com/lindawangg/COVID-Net (accessed on 19 December 2022). The work described in this paper is available at https://github.com/c-barakat/covidnet_tune (accessed on 19 December 2022).

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
