# Peer review of "Analysis of Chest X-ray for COVID-19 Diagnosis as a Use Case for an HPC-Enabled Data Analysis and Machine Learning Platform for Medical Diagnosis Support"

_diagnostics, 2023, doi:10.3390/diagnostics13030391_

Round 1

Reviewer 1 Report

The paper is well-written. However, the author should do the following minor corrections

1. The abstract section needs to be improved and concise. 

2.     The contributions of the paper should be listed pointwise in the introduction section.

3. The Conclusion section is missing in this manuscript.

4.      I had not seen any performance comparison of the proposed approach with other competing approaches. The author should mention why the proposed approach is not compared with other approaches. Otherwise, the author should compare with State-of-the-art and present the performance score in a Table.

Reviewer 2 Report

Article title: Analysis of Chest X-Ray as a Use Case for an HPC-enabled Data Analysis and Machine Learning Platform for Medical Diagnosis

Support

# Overall statement or summary of the article:    

The paper described the large-scale retraining of COVID-Net as a use case to validate a modular Medical Diagnosis Support Platform built on an HPC infrastructure and taking advantage of novel and efficient ML algorithms. The topic for this study is very interesting and sound, the research question is clearly outlined, and results are clearly presented. The proposed model is important in medical imaging technology; however, some points are required before any progress.

# It prefers to add “COVID-19” word in the title.

# Please add some of the most important quantitative results to the Abstract.

#   In section 1, there is a lack to link some information with references (e.g., this sentence “The standard approach using Reverse transcription polymerase chain reaction (RT-PCR) required a lot of time, trained staff, and laboratory capacity and showed, especially in the beginning of the pandemic, very heterogenous accuracy” needs a reference.

# In section 1, the authors should add a comparative overview table that shows the key differences between the different previous models and the proposed model.

# In section 2.3, it will be very useful to draw a flowchart or block diagram that describes the proposed model.

# Can you please add a section at the end of the manuscript “conclusion” to mention the usability of the proposed model, strategies or recommendations to reduce uncertainties in the study? the conclusion section should also contain what impacted the results, limitations and what is future work if any.

# The authors should write their paper in passive sentences, not in active to make it more academic (avoid using we).

Reviewer 3 Report

The authors has developed a deep learning-based platform for diagnosis support. While the results look promising, I have some doubts and hope authors could provide some clarifications.

1.     There are some typos and grammar errors in the writing. Please revise them. The format is not organized well, should be revised as well.

2. What is the pipeline for the platform? Is there any figure to illustrate this?

3. Some part is not easy to understand and authors should explain more. For example, what is the Hyperband module used for?

4. What is the core contribution in this paper? Is the HPC the core contribution? What is the difference between the proposed platform with some existing platforms?

5. Is there any privacy issue for this platform? 

6. Can authors provide the figure for the dataset? Some figures are not clear for people to read and understand. Please update.

7. Which dataset is used for evaluation step? What is the percentage to split the dataset for training, evaluation and testing? 

Round 2

Reviewer 2 Report

This version of the manuscript has been improved, and the authors have answered most asked questions. I recommend the acceptance of the paper for publication.

Reviewer 3 Report

The quality of this manuscript has been improved a lot.